# ChatGPT and L2 Written Communication: A Game-Changer or Just Another Tool?

**Artem Zadorozhnyy** * and **Wan Yee Winsy Lai**

Department of English Language Education, Education University of Hong Kong, 10 Lo Ping Road, Tai Po, New Territories, Hong Kong SAR, China; winsy.lwy@gmail.com
* Correspondence: archiezador@eduhk.hk

**Abstract:** The development of English language written communication skills across many contexts has been hindered by factors such as examination-oriented cultures, anxiety associated with both oral and written communication, limited opportunities for engaging in authentic communication, and a lack of individualized, personalized feedback. Multiple studies throughout the last decade have explored chatbot integration as a means to address these issues and enhance students' language learning toolkits. This paper delves into the potential benefits and roles of advanced Generative AI (GenAI) chatbots (e.g., ChatGPT) in enhancing second language (L2) communicative practices. We evaluate findings associated with various types of chatbots and present pedagogical strategies for their application of GenAI in both in-class and out-of-class spaces to support students' language learning experiences. We also propose future research directions, emphasizing the necessity to explore the ethical use of AI tools, their impact on L2 communication, and the comparative effectiveness of retrieval-based and GenAI-powered chatbots in language education.

**Keywords:** AI; ChatGPT; GenAI; language learning; language teaching; chatbots; computer-assisted language learning



## 1. Introduction

The development of English language communication skills continues to be a central focus for EFL (English as a Foreign Language) students[1] in educational settings across various contexts (Lee and Taylor 2022; González-Lloret 2022; Zadorozhnyy and Lee 2023). As digital technology advances, innovative tools essential for enhancing interactive methodologies in educational development effectively support the growth of both written and oral communication skills in EFL students (Kim et al. 2021; Hsu and Liu 2021). These tools enable students to negotiate meaning across various contexts and different audiences in another language. However, while the advancement of digital tools has been proven to be a significant asset in enhancing communicative competence, previous research acknowledges multiple factors embedded within traditional classroom activities that could potentially impede the progress of English language communication. For instance, studies have highlighted the emphasis of examination-oriented approaches leading to a lack of motivation (Fuchs 2019; Kiaer et al. 2021), anxiety related to oral and written communication (Liu and Ni 2015), limited opportunities to engage in authentic writing tasks (Sala-Bubaré and Castelló 2023), as well as a lack of individual and personalized feedback (Bakla 2020).

To address the associated issues, studies focus on the integration of chatbots as a means to help students overcome the underlined issues and enhance their language learning toolkit (Fryer et al. 2017; Guo et al. 2023; Kim et al. 2021; Kohnke 2022; Smutny and Schreiberova 2020). Research has demonstrated the beneficial effects of these tools for improving students' affective outcomes such as enjoyment (Kohnke 2023), motivation (Yin et al. 2021), confidence in learning English (Ebadi and Amini 2022), and willingness to communicate (Ayedoun et al. 2019), as well as linguistic outcomes that include listening

(Kim 2018), speaking performance (Lin and Mubarok 2021), and vocabulary (Alsadoon 2021). While the aforementioned studies integrating chatbots demonstrated promising results, the majority of this research utilized retrieval-based models, which present some limitations. Specifically, chatbots were often found to exhibit a limited capacity to generate original content, often produced inauthentic responses, were constrained by the availability of data, and struggled to comprehend multiple sentences simultaneously (Huang et al. 2022; Vinall and Hellmich 2022).

The advent of advanced AI platforms, such as ChatGPT, which employ more sophisticated algorithms, has the potential to significantly improve the language learning experience for EFL students (Godwin-Jones 2023; Huang et al. 2022). Responding to the increasing focus on AI-powered tools, recent publications highlight various approaches to integrating these tools and propose innovative strategies for their pedagogical implementation (Kohnke et al. 2023; Alharbi 2023). To expand pedagogical implications, this article aims to draw upon existing findings of empirical investigations and extensive reviews related to both retrieval-based chatbots and advanced AI-enabled tools, with a specific emphasis on offering pedagogical recommendations to enrich EFL students' language learning repertoire. Our goal is to illuminate a more diverse range of potential trajectories for developing second language (L2) communicative practices, leveraging the advancements of AI platforms.

To narrow down the scope of our article, we specifically concentrate on the development of written communication skills in a second language (L2) context, in line with the ACTFL World-Readiness Standards for Learning Languages (National Standards Collaborative Board 2015). The text-based nature of Generative AI (GenAI) models necessitates this focus. Given this, our primary focus is on equipping language educators with strategies to guide students' use of AI-powered tools within and beyond the classroom, thereby enriching their overall language learning experiences. Additionally, we propose potential future research directions, fostering ongoing dialogue on AI's effective use in language learning.

## 2. Chatbots and AI-Enhanced Language Learning Tools

As technology advances, chatbots are becoming an increasingly prominent tool in language education, enhancing learning experiences via interactive communication and personalized support (Fryer et al. 2019; Jia et al. 2012). The initial appearance of chatbots in educational contexts dates back to the 1970s when Weizenbaum (1966) pioneered the field of chatbots with the creation of ELIZA, an experimental attempt to facilitate human–computer interaction. Although ELIZA's capabilities were relatively limited, it paved the way for later, more advanced chatbots that can facilitate realistic dialogues with users (Godwin-Jones 2023). Over time, chatbot models have continued to evolve, leading to the current classification of these tools into two primary categories: retrieval-based and generative-based chatbots (Pandey and Sharma 2023). The retrieval-based chatbots function by selecting replies from a predefined database of conversation snippets (Han et al. 2021). On the other hand, generative-based chatbots (GenAI) employ machine learning techniques to autonomously create responses, providing a more dynamic and adaptable conversational experience (Mishra et al. 2023). The recent progress in AI-powered chatbots relies on natural language processing, which facilitates the transformation of textual information into structured data and allows for the extraction of valuable content (Godwin-Jones 2022). These "intelligent" mechanisms enable modern AI chatbots to possess the capacity to adapt and process unstructured user inputs while generating human-like natural language responses (Huang et al. 2022; Kim et al. 2021; Belda-Medina and Calvo-Ferrer 2022).

The development of GenAI components has advanced language learning tools and has prompted researchers to consider their efficacy in both in-class and out-of-class settings. Researchers focused on the integration of chatbots across multiple language learning areas, including grammar (De Gasperis and Florio 2012), vocabulary (Jia et al. 2012), as well as reading and listening skills (Kim 2018). Given the "intelligent" nature of GenAI, there has

been an increasing number of studies emphasizing L2 written communication skills as a central focus. These works highlight that students perceive chatbots as a comfortable, engaging, and enjoyable method to engage in communication activities and improve writing skills (Hill et al. 2015; Yin et al. 2021).

Previous research has highlighted that integrating chatbots can be beneficial in addressing *affective factors* related to L2 communication. Guo et al. (2023), for instance, introduced Argumate, a retrieval-based chatbot for argumentative writing, and discovered that students perceive the chatbot as providing a stress-free environment, which helped reduce anxiety. Students displayed positive attitudes and valued the support offered by the chatbot. The chatbot-generated content was found to stimulate thought, serve as a basis for engaging debates, and foster a relaxed learning atmosphere. Similarly, positive findings were stated by Ayedoun et al. (2015) who carried out a study with Japanese undergraduates and found that engaging with conversational agents can help reduce students' anxiety and increase their self-confidence in English communication. Additionally, engagement with conversational agents was found to foster a desire to use English in everyday conversation settings and effectively immerse learners in the conversational context. Similarly, positive outcomes were reported in relation to the development of intrinsic motivation, which serves as a precursor to students' engagement in chatbot-based micro-learning systems (Yin et al. 2021).

Similarly, the use of chatbots has demonstrated positive effects on the *behavioral outcomes* of EFL students. In a study conducted by Guo et al. (2023) involving Chinese undergraduate students in a public speaking course, chatbot-supported classroom debates were successfully implemented. The results showed that, concerning behavioral engagement, the chatbot promoted varied viewpoints and assisted students in exchanging, consolidating, and refining their ideas during debate preparation. In Kim (2016), students were categorized into low, medium, and high proficiency groups based on an English test. They were then randomly assigned to either an experimental group that interacted with a chatbot or a control group that communicated with human peers. The findings revealed that lower-level students engaged more in repetition and reformulation, while medium and high-level students sought clarifications more frequently compared to their counterparts in the control group. Regarding the duration of communication, Hill et al. (2015) reported that university students conversed with chatbots for noticeably longer periods of time than they did with real people. In a related study by Goda et al. (2014), the authors investigated the impact of chatbot assistance on group discussion preparation. The analysis revealed that the chatbot-assisted group engaged in more conversation activities during the group discussions, suggesting a positive effect of chatbot integration on student interactions.

Lastly, chatbots' impact has been noted in connection to *cognitive outcomes*, the domain-specific knowledge in students' learning performance, including vocabulary, writing, listening, and grammar (Huang et al. 2022). As such, Lee et al. (2015) integrated Genie Tutor, a hybrid system for correcting grammatical mistakes, which showed promising results in helping to correct students' writing and suggesting relevant vocabulary during communicative scenarios. Kim et al. (2019) found that Korean college students who interacted with the Replika chatbot outside of class exhibited improved grammar skills compared to those who conversed with human partners. The results revealed a significant improvement in grammar test scores for the chatbot group compared to the human partner group.

Overall, the increasing number of studies on chatbot integration reflects the exponential growth in chatbot development across the years, a trend corroborated by review studies in both general education (Okonkwo and Ade-Ibijola 2021) and language learning specifically (Huang et al. 2022, 2023). Another observable trend is the shift in research toward generative model chatbots that offer more dynamic and adaptable interactions (e.g., Dave the Debater by Le et al. 2018; ArgueBot by Wambsganss et al. 2021, among others). This trend emerges in response to the limitations observed in earlier chatbots, including issues related to content creation, inauthentic output, limited data availability, and the inability to understand multiple sentences at once (Huang et al. 2022; Vinall and Hellmich

2022). In light of these limitations, the introduction of AI-powered generative models, such as ChatGPT-3 and ChatGPT-4 by OpenAI, has significant potential for enhancing L2 written communication practices.

### 3. ChatGPT: Introduction and Implications for Practice

The arrival of advanced AI-powered tools, such as ChatGPT, has been at the forefront of debates and discussions since its launch in November 2022. ChatGPT, one of the most advanced AI-powered chatbots, was developed by OpenAI, a Microsoft-backed company, and the first iteration of this model, GPT-1, was introduced in June 2018 (Ray 2023). The advancement of AI-driven chatbots is fundamentally based on the usage of natural language processing (NLP), machine learning (ML), and deep learning (DL) mechanisms (Ray 2023; Mishra et al. 2023; Godwin-Jones 2023). With the use of these technologies, advanced chatbots may respond to user inquiries by extensively accessing vast language databases (Godwin-Jones 2023; Huang et al. 2023).

The implementation of innovative technology has increasingly captured the attention of SLA researchers (Huang et al. 2023; Hwang and Chien 2022; Salvagno et al. 2023). Educators emphasize the benefits of such AI tools in fostering L2 communicative proficiency, and the interest in their potential continues to grow. Previous research has emphasized the capabilities of AI-enabled chatbots, which include offering immediate feedback, personalization, and customization opportunities, delivering linguistic input and facilitating daily conversational practice, as well as fostering comprehension of language features (Ayedoun et al. 2019; Fryer et al. 2017; Guo et al. 2022; Wang et al. 2020). Given the launch of advanced state-of-the-art AI tools, there are compelling reasons to outline potential directions for ChatGPT's application in L2 acquisition, specifically targeting the enhancement of L2 written communication skills.

To start with, GenAI can serve as a tutor to assist students when they encounter difficulties understanding and clarifying content. Often, language learners' proficiency may not be sufficient to comprehend advanced language structures (e.g., tenses, grammatical features) in authentic texts. In these instances, GenAI could be employed to simplify the content and act as a supportive companion, enriching teaching activities by providing answers to questions or offering supplementary explanations (see Figure 1).

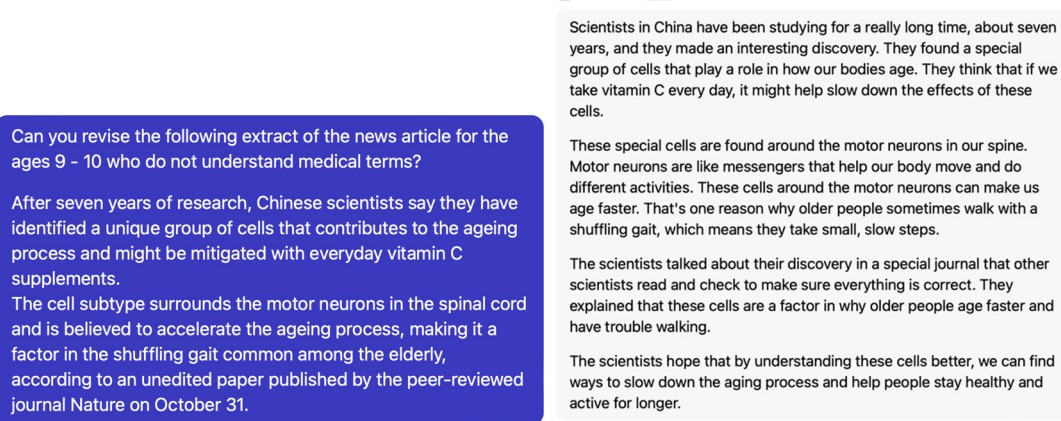

**Figure 1.** ChatGPT-supported revision to simplify the content.

Earlier, Kohnke et al. (2023) noted that ChatGPT can adjust the complexity of generated content, allowing for differentiation in materials to suit students at various levels. This feature sets ChatGPT apart from traditional conversational agents and scripted dialogue systems, which are limited by the topics pre-programmed by their creators.

Proceeding further, ChatGPT can be employed for role-playing conversations, simulating diverse characters and situations. As highlighted by Fryer et al. (2019), chatbot affordances can be effectively used for simulation activities. Leveraging the advanced

capabilities of ChatGPT algorithms could provide language learners with opportunities to practice conversational skills and enhance fluency in an immersive and engaging manner in comparison with retrieval-based chatbots (see Figure 2). Building on this, prompted GenAI platforms can help students familiarize themselves with different communication genres (i.e., formal, semi-formal, and informal) in a consistent, scaffolded manner. This can be achieved via various scenarios such as customer service interactions, travel agency role-plays, and job interview practice, all of which can further enhance their understanding and application of newly acquired language skills.

**Figure 2.** ChatGPT-supported role-play.

When properly directed, ChatGPT can serve as an engaging game companion. For instance, educators can instruct students to use GenAI tools to play a "Guess the object/word" game to reinforce learners' linguistic competencies, such as vocabulary acquisition, sentence construction, and oral and written descriptive ability. In this setup, students can practice communicating about different attributes and characteristics of an object while ChatGPT makes an attempt to determine what it is. Alternatively, learners can participate in AI-assisted *Collaborative Storytelling* activities when GenAI and students contribute to a shared narrative, allowing learners to refine their storytelling abilities, vocabulary application, and creativity (see Figure 3).

Another promising avenue for GenAI integration is utilizing platforms like ChatGPT-3 as an AI-enabled "mentor". Students could input their writing and receive guidance on enhancing sentence structure, grammar, and spelling (see Figure 4). Prior studies have highlighted the utility of chatbots in providing prompt feedback on students' written work, identifying errors, and offering suggestions for improvement (Ranalli and Yamashita 2022; Ayedoun et al. 2019). AI-powered tools can elevate this feedback process by delivering personalized learning strategies and resources. For example, AI tools can recommend relevant language learning applications tailored to the specific skills a learner aims to improve (see Figure 5). However, the full potential of cutting-edge GPT models in this context remains to be explored.

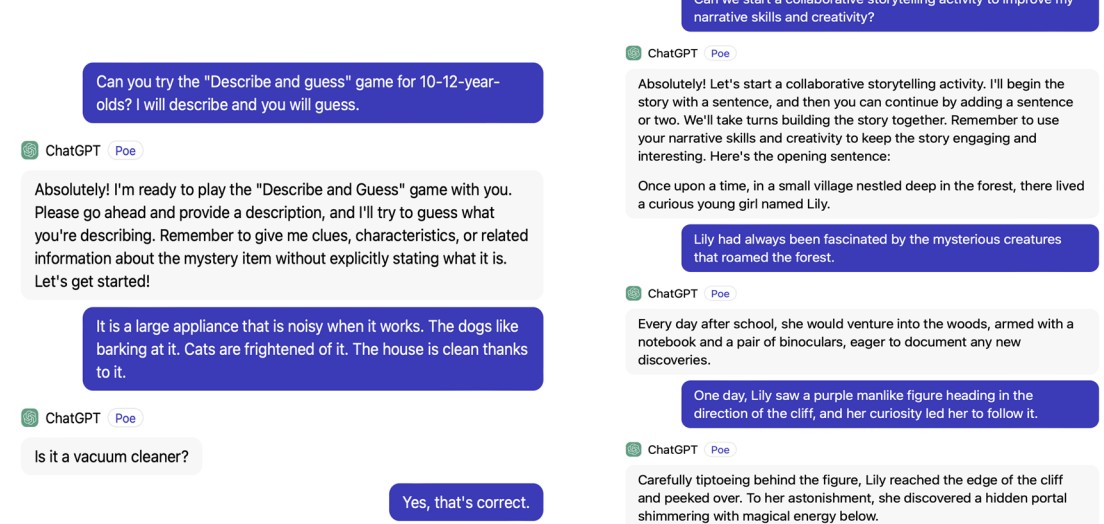

**Figure 3.** ChatGPT-supported "Guess the object" game and collaborative storytelling activity.

Can you proofread and explain the grammar and spelling mistakes (if any) on the paragraph below on "A day of a fireman".
Last weak, the fireman's day starts with a loud alam. He quickly gets dressed and rushe out to the fire station. There's no time to waste when a fire breaks out! He jumbs into the firetruck, sirens blaring, and drives fast to the scene. When they arrive, the firemen works together to put out the flames. They use hosess, axes, and other special tools to extinguish the fire. Sometimes, they even have to rescue people from burning buildings. It's a dangerus job, but the fireman is brave and proud to serve his commnuity. After a long day of fighting fires, he returns to the fire station, tired but satisfied. He knows he has helped people and made a diffrence. Being a fireman is a challenging but rewardin profession.

Proofread Version:
Last week, the fireman's day starts with a loud alarm. He quickly gets dressed and rushes out to the fire station. There's no time to waste when a fire breaks out! He jumps into the firetruck, sirens blaring, and drives fast to the scene. When they arrive, the firemen work together to put out the flames. They use hoses, axes, and other special tools to extinguish the fire. Sometimes, they even have to rescue people from burning buildings. It's a dangerous job, but the fireman is brave and proud to serve his community. After a long day of fighting fires, he returns to the fire station, tired but satisfied. He knows he has helped people and made a difference. Being a fireman is a challenging but rewarding profession.

Explanation of Corrections:

1. "Last weak" should be "Last week" (spelling mistake).
2. "alam" should be "alarm" (spelling mistake).
3. "rushe" should be "rushes" (spelling mistake).
4. "jumbs" should be "jumps" (spelling mistake).
5. "firetruck" should be "fire truck" (missing space).
6. "firemen" should be "firefighters" or "firemen work" (agreement error).
7. "hosess" should be "hoses" (spelling mistake).
8. "dangerus" should be "dangerous" (spelling mistake).
9. "commnuity" should be "community" (spelling mistake).
10. "diffrence" should be "difference" (spelling mistake).
11. "rewardin" should be "rewarding" (spelling mistake).

**Figure 4.** ChatGPT-supported proofreading and explanation on written input.

Lastly, utilizing advanced ChatGPT-like models can simultaneously bolster L2 writing communication skills and reading comprehension. By generating personalized short texts or dialogues, AI models provide a rich, engaging context for practicing written communication while also enhancing reading abilities. GenAI can equip learners with a wide range of vocabularies, idioms, and colloquial expressions, thereby expanding their vocabulary bank for potential scenarios in practicing communication and enhancing their understanding of its contextual use. Beyond this, GenAI can extend its utility by offering follow-up questions for comprehension checks and verifying learners' answers.

Summing up, ChatGPT's ability to offer tailored recommendations and practices underscores the broader potential of AI tools as invaluable assets in language learning. Suggested ideas are limited by the scope of the paper, but nonetheless, educators should scaffold students' understanding of the nuances involved in working with AI-based tools in in-class spaces and beyond. The potential of AI tools can be effectively demonstrated in class, with educators guiding students in formulating effective prompts for communication with chatbots. For assignments outside the classroom, educators can provide initial prompts

to initiate students' interactions with the AI. Encouraging students to create their own prompts for chatbot interactions not only boosts engagement but also aids in steering the conversations. Subsequently, these interaction logs could serve as review material or the foundation for reflective essays, fostering a progressive approach to learning. Collectively, these experiences could become a component in bridging language learning practices across formal and informal online learning spaces, thus establishing a consistent, valuable learning continuum.

**Figure 5.** ChatGPT-generated suggestions of pronunciation-centered mobile applications.

## 4. Future Research Directions

In the review of AI-enhanced language learning from 2000 to 2019, Huang et al. (2023) reported that almost a third of all academic publications focused on automated writing evaluation and computer-mediated communication with chatbots. Similarly, previous research has demonstrated the potential benefits of chatbots for students (Fryer et al. 2019; Huang et al. 2023; Guo et al. 2022); however, many of these studies utilized chatbots with constrained capabilities. The advent of more advanced platforms, such as ChatGPT and other GenAI technologies, could substantially enhance these benefits. The strategies suggested for incorporating ChatGPT and related tools into students' language learning repertoire offer valuable insights into future instructional uses; nonetheless, they are merely the beginning. GenAI technologies provide the foundation for more diversified research agendas related to the development of L2 communication skills among EFL learners.

One important area for future research could be investigating strategies for fostering responsible and ethical use of these tools among students. With the emergence of new advanced AI-powered platforms in language education, researchers suggest placing a greater focus on the issues of plagiarism and cheating (Kohnke et al. 2023). AI algorithms are at the core of multiple tools such as automated written corrective feedback (AWCF) platforms (Ranalli and Yamashita 2022), machine translation (MT) tools (Lee 2023), and AI-assisted automatic text completion and alternative phrasing suggestions (Dale 2020; Godwin-Jones 2022). The critical question that surfaces from this lack of awareness is how to differentiate content generated by AI tools from human-produced work. Hence, it is imperative for educators to raise students' awareness about the AI engines that drive these tools in order to promote academic honesty and maintain scholarly integrity. By equipping students with a comprehensive understanding of AI-generated content and its implications, educators can enable learners to engage in logical, analytical, and ethical practices throughout their advanced academic pursuits. Owing to this, future studies could

explore ways to integrate AI tools seamlessly into the learning process by adapting and possibly revising relevant theoretical frameworks.

Another research direction is the potential impact of a metaverse environment for L2 communication brought by generative AI tools. Since all interactions are recorded in the metaverse, "avatars" can learn from the prior conversation of the knowledge of the individuals to conduct an unscripted conversation with learners, or learners can communicate with other L2 speakers who are present virtually in the metaverse and incidental learning (Godwin-Jones 2023). This new space for L2 communication provides a language partner with personal knowledge of the user and infinite patience and availability for learners. Building upon these advancements, it is important to note that Chat GPT is set to introduce an aural/oral aspect. This upcoming development will not only enhance its current capabilities but also unveil new potential in language learning, adding a fascinating dimension to future research.

At last, while previous studies have underscored behavioral patterns associated with engagement involving chatbots, there is room to extend this line of inquiry. Fryer et al. (2017) reported that the efficacy of chatbots in promoting human–computer interactions diminished over time, while students' engagement with human partners remained consistently elevated during the execution of EFL speaking tasks. A comparative analysis of students' perceptions of L2 communication involving retrieval-based chatbots and advanced AI-powered tools, such as ChatGPT, warrants further investigation. It might be of interest to examine the persistence of the novelty effect and the distinctions in perception among students at varying educational stages (e.g., secondary and high school) with respect to such interactions. Understanding if students can distinguish between these platforms and recognize their impact on language learning is another important area to explore.

## 5. Conclusions

The emergence of artificial intelligence (AI) technology, particularly the advent of ChatGPT, has opened up new possibilities for innovative language learning practices. The ability of AI platforms to provide instant feedback, personalization, and authentic communication scenarios has the potential to significantly impact the development of L2 communication. Moving forward, the integration of ChatGPT into various applications and communication platforms will continue to revolutionize language learning. As the AI field advances, it is only a matter of time before ChatGPT and similar technologies are integrated into more diverse and interactive learning environments. Therefore, educators are advised to take advantage of new emerging opportunities related to AI-enabled platforms and teach students the necessary skills to engage with AI tools in a responsible and effective manner while critically evaluating AI-generated outputs.

**Author Contributions:** Conceptualization, A.Z. and W.Y.W.L.; methodology, A.Z. and W.Y.W.L.; software, A.Z. and W.Y.W.L.; formal analysis, A.Z. and W.Y.W.L.; investigation, A.Z. and W.Y.W.L.; writing—original draft preparation, A.Z. and W.Y.W.L.; writing—review and editing, A.Z. and W.Y.W.L.; visualization, A.Z. and W.Y.W.L. All authors have read and agreed to the published version of the manuscript.

**Funding:** The work is a part of research funded by the Hong Kong Postdoctoral Fellowship Scheme (Grant number PDFS2223-8H04).

**Institutional Review Board Statement:** Not applicable.

**Informed Consent Statement:** Not applicable.

**Data Availability Statement:** Not applicable.

**Acknowledgments:** During the development of this work, the authors used the AI service provided by Poe's GPT-4 platform (https://poe.com/GPT-4, accessed on 12 November 2023) for proofreading services, ensuring linguistic consistency of thoughts, and improving the document's narrative flow. The screenshots provided as illustrations were made while using the aforementioned Poe's GPT-4 and OpenAI's ChatGPT-3. The input provided by the platform was subsequently subjected to a

thorough review and necessary modifications by the authors, who take full responsibility for the content of the publication.

**Conflicts of Interest:** The authors declare no conflict of interest.

## Notes

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
