# Peer review of "ChatGPT and L2 Written Communication: A Game-Changer or Just Another Tool?"

_languages, doi:10.3390/languages9010005_

Round 1

Reviewer 1 Report

Comments and Suggestions for Authors

The paper discusses the significance of considering both potential and challenges when implementing advanced AI-powered tools, like ChatGPT, in language teaching and learning. However, the overall goal of the paper requires further clarification. Is it intended to be a pedagogical piece or a literature review?

To strengthen the overall quality of the paper, thorough editing is needed to address grammar issues, misspellings, coherence, punctuation, and referencing techniques.

 The author(s) state(s): [] we will explore how AI-enabled chatbots and engines can be incorporated into the development of second language (L2) communicative practices. Pedagogical tips will be provided on the relevance of their application for EFL teaching and learning, emphasizing the impact on students' language learning experiences in both in-class and out-of-class contexts and enriching their overall language learning repertoires. At last, we will suggest potential research directions for future studies on this topic.

However, since no actual data has been collected, the suggested pedagogical tips should be presented more critically.

If the suggested research directions are based on the literature review (given that no data was collected), the literature review section of the paper should receive more emphasis and thorough methodological attention. For instance, there could be more transparency in relation to the inclusion/exclusion criteria for the research synthesis, presented as a brief outline in the introduction section. Additionally, the authors should clearly state the databases that were searched, the total number of empirical studies included in the analysis, and other relevant details.

 Regarding Table 1, it is not properly integrated with the rest of the text. To address this, the table should be properly introduced and its relevance to the discussion should be explained.

 Overall, by addressing these issues, the paper will be improved in clarity and quality, making it more effective in conveying its message.

Comments on the Quality of English Language

The paper would benefit from thorough editing to address grammar issues, misspellings, coherence, punctuation, and referencing techniques.

Author Response

Dear Reviewer, 

Our heartfelt thanks go to you for your time and constructive comments. We have perused all the comments carefully and revised the manuscript accordingly. Please see our responses below. Thank you very much.

Best Regards, 
Authors

Reviewer 2 Report

Comments and Suggestions for Authors

Methods and conclusions should be developed.

Although there are many studies on artificial intelligence-based ChapGpt, the literature has not been adequately examined.

Author Response

(The authors gave the same response as above.)

Reviewer 3 Report

Comments and Suggestions for Authors

Your review topic offers a timely consideration of the positive contributions ChatGPT could make to support language learning. This is a worthy endeavor. However, the constraints you yourself mention regarding the length restrictions have impacted the potential usefulness of your contribution. It is so compressed that the potential reader would have to actually review many of the references you sight to fully appreciate the points you are making. 

Also, there is little in the article that is limited to or specific to the Asian context. Most of what you were saying would be relevant to any language learner anywhere.

I also had a few questions as I read:

page 2, line 85: How can “chatbot-generated content “present an engaging debate method?”  A method might dictate HOW to use a text but does not constitute a method in itself.

Page5,  Line 188 Chatbot and roleplay

“In a MORE immersive and engaging manner”.. More than WHAT?

 Line 189

When you say that chatGPT can offer examples in alternative dialects, how do you define dialects? (How 'dialects' are defined is a complicated issue. e.g. While Chinese fangyan are often referred to in English as dialects, for many linguists and native English speakers, a number of these would be mutually untintelligible languages, each of which could have mutually comprehensible varieties.)

Page5,  Line 188 Chatbot and roleplay

“In a MORE immersive and engaging manner”.. More than WHAT?

Comments on the Quality of English Language

I have noticed just a few local errors in English usage, which is excellent.

Pay attention to the discourse level usage of the definite article. which sometimes appears where the specifics are not yet known to the reader.

Author Response

Dear Reviewer, 

Our heartfelt thanks go to you for your time and constructive comments. We have followed all the comments carefully and revised the manuscript accordingly. Please see our responses below. Thank you very much for your consideration.

Best Regards, 
Authors

Reviewer 4 Report

Comments and Suggestions for Authors

For coherence, add numbers to ALL the subtitles. 

Lines 28-20: Explain what you mean by the acronym EFL. This is important because you begin the paragraph by referring to English language communication (not English as a Foreign Languages), and this may confuse your reader. 

Lines 41-42: Add ‘at’: “…multiple sentences at once.”

Line 51: Again, explain the meaning of EFL in “… EFL teaching and learning…” and how does it relate to “second language (L2) communicative practices” above. Alternatively, you can explain at the beginning of the article that these two concepts will be used interchangeably.

Line 74: Why open the sentence with ‘however’?

Line 75: Again, please define “L2 communication skills.” For instance, are you only including oral communication skills, which include listening skills. But it seems that you are including writing skills in your definition of L2 communication skills as well. Please, clarify. One possibility could be to follow the ACTFL World Readiness Standards for Learning Languages.

In figure one: Capitalize the first word “please” in the blue box.

Line 198: Add ‘can’ or ‘may’: “Similarly, it (can/may) be used for homework support…”

Line 212: change the ‘n’ in ‘nonetheless’ to lowercase “but nonetheless, educators must scaffold…” 

Line 221: I suggest changing “Huang and colleagues” for Huang et al.  

Line 230: in (Lee, 2023), is 2023 the page or the year? This is not clear, please correct.

Line 239: see error in “ethnical usage of AI tools…” change ethnical to ethical

Please, review citation, be coherent. 

Author Response

Dear Reviewer, 

Our heartfelt thanks go to you for your time and constructive comments. We have perused all the comments carefully and revised the manuscript accordingly. Please see our responses below. Thank you very much for your consideration.

Best Regards, 
Authors

Round 2

Reviewer 1 Report

Comments and Suggestions for Authors

The authors revised the manuscript according to the comments and suggestions provided by the reviewer. The paper now reads as a more coherent and compelling piece of writing.

Author Response

Thank you for your insightful comments and suggestions. Your constructive feedback has greatly helped in refining our manuscript. We believe that your suggestions have enhanced the coherence and persuasiveness of the paper, allowing it to resonate more effectively with readers.

Reviewer 3 Report

Comments and Suggestions for Authors

The current revised version of the paper addresses all previous concerns and should make an excellent addition to the conversation on using AI/Chat GPT to support written language learning with writing in alternative and interactive digital genres. The final version might add that Chat GPT is in the process of developing an aural/oral aspect available shortly.

Author Response

Thank you for your insightful comments and positive feedback on our revised paper. We appreciate your suggestion regarding the ongoing development of an aural/oral aspect in Chat GPT.

In response, we have included information about this upcoming feature in the final version of the manuscript. 

Once again, thank you for your valuable contribution to enhancing the quality of our work.